# Genetic Analysis of HIBM Myopathy-Specific GNE V727M Hotspot Mutation Identifies a Novel COL6A3 Allied Gene Signature That Is Also Deregulated in Multiple Neuromuscular Diseases and Myopathies

**DOI:** 10.3390/genes14030567

**Published:** 2023-02-24

**Authors:** Shivangi Attri, Moien Lone, Amit Katiyar, Vikas Sharma, Vinay Kumar, Chaitenya Verma, Suresh Kumar Gahlawat

**Affiliations:** 1Department of Biotechnology, Chaudhary Devi Lal University, Sirsa 125055, India; 2Department of Biochemistry, All India Institute of Medical Sciences, New Delhi 110029, India; 3Centralized Core Research Facility, All India Institute of Medical Sciences, New Delhi 110029, India; 4The Dorothy M. Davis Heart & Lung Research Institute, Department of Internal Medicine, The Ohio State University Wexner Medical Center, Columbus, OH 43210, USA; 5Department of Physiology and Cell Biology, Department of Internal Medicine, The Ohio State University Wexner Medical Center, Columbus, OH 43210, USA; 6Department of Pathology, The Ohio State University Wexner Medical Center, Columbus, OH 43210, USA

**Keywords:** myopathy, GNE, ncRNA, HIBM, V727M, COL6A3, mutation, neuromuscular disorders

## Abstract

The GNE-associated V727M mutation is one of the most prevalent ethnic founder mutations in the Asian HIBM cohort; however, its role in inducing disease phenotype remains largely elusive. In this study, the function of this hotspot mutation was profoundly investigated. For this, V727M mutation-specific altered expression profile and potential networks were explored. The relevant muscular disorder-specific in vivo studies and patient data were further analyzed, and the key altered molecular pathways were identified. Our study found that the GNEV727M mutation resulted in a deregulated lincRNA profile, the majority of which (91%) were associated with a down-regulation trend. Further, in silico analysis of associated targets showed their active role in regulating Wnt, TGF-β, and apoptotic signaling. Interestingly, COL6a3 was found as a key target of these lincRNAs. Further, GSEA analysis showed HIBM patients with variable COL6A3 transcript levels have significant alteration in many critical pathways, including epithelial-mesenchymal-transition, myogenesis, and apoptotic signaling. Interestingly, 12 of the COL6A3 coexpressed genes also showed a similar altered expression profile in HIBM. A similar altered trend in COL6A3 and coexpressed genes were found in in vivo HIBM disease models as well as in multiple other skeletal disorders. Thus, the COL6A3-specific 13 gene signature seems to be altered in multiple muscular disorders. Such deregulation could play a pivotal role in regulating many critical processes such as extracellular matrix organization, cell adhesion, and skeletal muscle development. Thus, investigating this novel COL6A3-specific 13 gene signature provides valuable information for understanding the molecular cause of HIBM and may also pave the way for better diagnosis and effective therapeutic strategies for many muscular disorders.

## 1. Background

Hereditary inclusion body myopathy (HIBM) represents an adult-onset muscle disorder characterized by typical histopathology findings on muscle biopsy along with severe progressive muscle weakness [1]. Cases of HIBM have been described worldwide, including in China, India, Japan, and Africa; however, this myopathy continues to be clinically and translationally under-recognized and poorly diagnosed [2,3,4,5]. Such patients are generally characterized by progressive proximal and distal weakness and wasting of both upper and lower limbs, usually beginning after 20 years of age, thus making them wheelchair-bound [6]. This form of myopathy results from mutations in the GNE gene present on chromosome 9p13-p12, which codes for a bifunctional enzyme (UDP-GlcNac epimerase/ManNac Kinase; GNE/MNK). This enzyme catalyzes a rate-limiting step during the biosynthesis of sialic acid (5-N-acetylneuraminic acid), found on glycoconjugates as terminal sugar [7,8]. The N-terminal portion of GNE/MNK (UDP-GlyNAc 2-epimerase) catalyzes epimerization of UDP-GlyNAc to ManNAc. The C-terminal fragment (ManNAc kinase) of the GNE gene phosphorylates ManNAc to ManNAc-6-P and phosphoenolpyruvate. ManNAc-6-P is then further condensed to sialic acid, which is next used as a substrate for the sialylation of glycoconjugates in the Golgi complex by sialyltransferases. It is believed that decreased production of sialic acid due to mutations in the GNE gene could result in muscle degeneration in HIBM; however, the exact cellular mechanism resulting in muscle degeneration remains largely unexplored [6,9,10,11]. Thus, the identification of new players is essentially needed to better understand the biological cause of the disease.

Long intergenic noncoding RNA (lincRNA) represents a significant group of recently identified noncoding RNA. They range in size from several hundred to tens of thousands of bases (≥200). Unlike other known lncRNAs (e.g., Antisense, intronic, or bidirectional), lincRNAs are located between protein-coding genes along with separate transcriptional units [12,13,14]. Although thousands of human lincRNAs have been identified, only a small percentage have been characterized. Recent findings suggest these lincRNAs could emerge as a new regulatory molecule, exemplified by their cell type-specific expression and subcellular compartment localization. This group of RNAs has also been partially correlated with pathogenic mechanisms, which suggests their role in various diseases. In one such study, lincRNAs-p21 was associated with tumor development and progression by functioning as a critical repressor of TP53 by interacting with ING1b and MDM2 proteins [15]. In diseases like rheumatoid arthritis (RA), lincRNA-p21 was also found to be under-expressed in blood samples, which were increased by the treatment of Methotrexate (MTX). Another study by Spurlock et al. showed that MTX modulates NF-κB activity through the induction of lincRNA-p21, suggesting a role of lincRNA-p21 in regulating NF-κB activity in RA [16]. Similarly, systemic lupus erythematosus (SLE) specific organ damage is associated with linc0949 alterations. The expression of these lincRNAs is not only significantly decreased in PBMCs of patients with organ damage, but linc0949 expression was found to increase after treatment. Further, it has been established as a potential biomarker for diagnosis and assessing disease activity, organ damage, and therapeutic responses in SLE [17]. Indeed, the association of several of the other well-described lincRNAs, such as HOTAIR, Xist, lincRNA-p21, and MALAT-1, with various diseases suggests lincRNAs are essential factors in the development and progression of cancer [18,19,20] and other diseases [21,22,23,24]. Although multiple conditions have been found to correlate with lincRNAs, the role of this ncRNA group remains unexplored in myopathies. 

The present study focused on identifying the lincRNAs associated with altered profiles in myopathies, with special emphasis on HIBM myopathy. For this, we induced overexpression of GNE-specific V727M mutation in in vitro models and explored altered noncoding RNA profiles for HIBM myopathy. Our analysis showed multiple lincRNA alterations that regulated genes involved in Wnt, TGF-β, and apoptotic signaling pathways. Further analysis of downstream targets found 13 lincRNA-associated genes to be altered, including COL6a3 as one of the highly overexpressed V727M mutation-specific targets. Interestingly, analysis of COL6A3 correlated genes found 12 of the coexpressed genes also to be altered by induction of the GNE V727M mutation, and all of these were shown to be having an upregulated trend. A similar upregulated expression of most of these genes was found in in vivo models as well as in many other muscular disorders. Thus, analysis of GNE-specific V727M mutation led to the identification of the novel COL6a3-specific 13 gene signature, altered in many musculoskeletal disorders. These genes may affect critical pathways linked to myopathies, including extracellular matrix organization, cell adhesion, and Skeletal muscle development, thereby leading to myopathy.

## 2. Materials and Methods

### 2.1. Cell Culture

HEK293 cells were maintained as per guidelines provided by American Type cell culture (ATCC). Briefly, human embryonic kidney (HEK293) cells were cultured in a CO_2_ incubator (M/S Thermo) at 37 °C and 5% CO_2_. Cells were grown in 75 mL culture flasks containing 10 mL Modified Eagle medium (MEM; Sigma) and subcultured every four days.

### 2.2. GNE Gene Cloning 

For cloning, the CDS region of the human GNE gene was PCR amplified using a primer having restriction sites for HindIII and BamHI in forward and reverse primers (Appendix A). The amplified product was run on 1.5% agarose gel for separation. The single band size showing GNE product size was cut, and DNA was purified using the QIAquick Gel Extraction Kit (M/s Qiagen). The PCDNA3.1+ vector and purified product were digested using restriction enzymes HindIII and BamHI and were ligated overnight at 4 °C. The amplified product was used for Transformation on competent cells that were next grown overnight in LB Agar media using Ampicillin as a selection marker. The single colonies grown after 14 h were used to inoculate LB media and later isolate DNA. The validation of the CDS insert was confirmed using Sanger sequencing. 

### 2.3. Site-Directed Mutagenesis (SDM)

For introducing V727M mutation-specific G to A mutation in the GNE gene, two SDM primers were designed that contained the desired nucleotide change (Appendix A). For SDM, 1 ng of plasmid DNA containing a wild-type GNE CDS region was denatured and amplified using the primers containing the desired mutation using Pfu turbo Polymerase in a final reaction volume of 25 µL and amplified for 30 PCR cycles. This enzyme uses non-strand-displacing action to extend and incorporate the mutagenic primers, resulting in nicked circular strands. The methylated, non-mutated parental DNA template was digested with the DPNI enzyme. The plasmid strand was next transformed into competent cells, where the cells repair the nicks in the mutated plasmid. The single colony was screened for the mutant GNE gene harboring V727M mutation. The validation of site-directed mutation was confirmed using Sanger sequencing. 

### 2.4. Plasmid Extraction and Purification

Various plasmids were transformed to DH5α competent cells and amplified in LB medium containing specific antibiotics overnight using a 37 °C shaker with a speed of 225 rpm. Then, the bacteria were harvested, and plasmids were extracted and purified using a Qiagen plasmid extraction and purification kit according to the procedure provided by the manufacturer.

### 2.5. Transfection

HEK293 cells were grown in Dulbecco’s Modified Eagle’s medium (DMEM) supplemented with 10% fetal bovine serum with 1% antibiotics and maintained in a tissue culture incubator at 37 °C with 5% CO_2_. Cells were transfected with GNE V727M overexpressing plasmid (PC-GNE-V727M), GNE wild type overexpressing plasmid (PC-GNE wt), or its control, PCDNA (3 µg for 6-well plate format), using Lipofectamine-2000 transfection reagent (M/s Invitrogen, Waltham, MA, USA) as per the manufacturer’s instructions. The cells were assayed after 48 h for quantification of transcript levels using qPCR and downstream RNAseq studies.

### 2.6. Total RNA Isolation

Total RNA isolation from cell lines from transfected cells was performed using an RNA extraction kit (M/s Qiagen, Hilden, Germany) according to the manufacturer’s instructions. The concentration and purity of RNA samples were determined using a NanoDrop spectrophotometer (M/s Thermo Fisher Scientific, Waltham, MA, USA).

### 2.7. Reverse Transcription and Real-Time PCR Studies

GNE levels were evaluated in HEK293 cells transfected with pcDNA3.1 (+) plasmid containing wild-type GNE (wt), mutant V727M (mut) GNE forms, or an empty vector. For each sample, 500 ng of isolated RNA was used for cDNA synthesis using a Superscript-vilo cDNA synthesis kit (M/s Thermofisher, Waltham, MA, USA). Expression for GNE transcripts as well as validation of other targets was done using gene-specific primers and SYBRGreen-based chemistry (Appendix A). GAPDH was used as a reference for normalizing gene expression data. Finally, fold change for each gene analyzed was calculated using the 2−ΔΔCt method. 

### 2.8. RNAseq Study 

HEK293 cells with successful transient transfection of GNE wt and mut forms (as validated by qPCR studies, Appendix A) were next taken for RNAseq studies. RNA extracted from HEK293 cells successfully overexpressing GNE wt and Mut forms were used for next-generation sequencing studies. For this, the biological replicates from two separate experiments for each (GNE wt and Mut) were used for RNAseq studies. Briefly, the extracted RNA for all four samples was quantified using Qubit RNA HS Assay (Invitrogen, Cat# Q32855). RNA purity was checked using QIAxpert, and the RNA integrity was assessed on TapeStation using High Sensitivity RNA screenTapes (Agilent, Cat# 5067-5579). A NEB Ultra II directional RNA-Seq Library Prep kit (NEB, Cat# E7760L) was used to prepare libraries for total RNA sequencing. First, the ribosomal RNA (rRNA), which constitutes ~95% of the total RNA population, was removed from 100 ng of total RNA using biotinylated, target-specific oligos combined with Ribo-Cop rRNA removal beads (Lexogen, Cat#144). Following purification, the ribs-depleted RNA was fragmented using divalent cations under elevated temperatures. The cleaved RNA fragments were copied into first-strand cDNA using reverse transcriptase. Second-strand cDNA synthesis was performed using DNA Polymerase I and RNase H enzyme. The cDNA fragments were then subjected to a series of enzymatic steps which repaired the ends and tailed the 3′ end with a single ‘A’ base, followed by ligation of the adapters. The adapter-ligated products were then purified and enriched using the following thermal conditions: initial denaturation of 98 °C for 30 sec; 13 cycles of −98 °C for 10 s, 65 °C for 75 s; final extension of 65 °C for 5 min. PCR-enriched libraries were then purified and checked for fragment size distribution on a Fragment Analyzer using a HS NGS Fragment Kit (1–6000 bp) (Agilent, Cat# DNF-474-1000). The libraries were quantified using Qubit High Sensitivity Assay (Invitrogen, Cat# Q32854). The quantified libraries were pooled and diluted to the final optimal loading concentration before cluster amplification on the Illumina flow cell. Once the cluster generation was completed, the clustered flow cell was loaded on the Illumina Novaseq 6000 instrument to generate 60 M, 150 bp paired-end reads.

### 2.9. Data Processing and Quality Control

The sequenced RAW data was subjected to Quality Control using FastQC. Subsequently, data analyses were performed in silico. Biological replicate samples were grouped in GNE WT or Mut for data analysis. The raw reads were next filtered using Trimmomatic (v-0.36) for quality scores and adapters. Contamination removal was performed using Bowtie2 (2.2.4). Filtered reads were aligned to the human genome (hg19) using spliceaware aligners such as HISAT2 (2.1.0) to quantify reads mapped to each transcript. The raw read counts were estimated using FeatureCount (1.5.2). The alignment percentage of reads ranged between 96 and 97% for all the samples. The total number of uniquely mapped reads was counted using feature counts. Finally, the uniquely mapped reads were subjected to differential gene expression using DeSeq2. 

### 2.10. Differential Expression Analysis

The raw read counts were normalized using DESeq2. The ratio of normalized read counts for mutant versus wild type (i.e., Mut/wild type fold change) was established, and the final data were represented as log2 fold change. Genes were first filtered based on the p-value (≤0.05). The distribution of these log2 (fold change) values was found to be normal. Those genes that were found to have −1 ≤ log2 (fold change) ≥ 1 were considered statistically significant.

### 2.11. LincRNA Target Prediction 

The potential targets of lincRNAs were predicted using rtool software (http://rtools.cbrc.jp/cgi-bin/RNARNA/index.pl accessed on 6 May 2021). A maximum of 400 genes predicted based on minimum energy criteria were considered important ones. For obtaining robust lincRNA regulated molecules, identified targets common to at least 4 out of 46 lincRNAs were considered for downstream analysis. These were next used for transcription factor analysis, GO biological pathway analysis, GO cellular component analysis, and Panther pathway analysis using the Genecodis tool. Finally, coexpressed gene analysis was performed using “genelikeme” (https://genecodis.genyo.es/ accessed on 26 May 2021).

### 2.12. Gene Set Enrichment Analysis (GSEA) 

For gene set enrichment analysis (GSEA), Hereditary Inclusion Body Myopathy (HIBM) expression profile data (GSE12648) were used. This study included expression profiling from muscle specimens from 10 HIBM patients, all of whom were carrying the M712T mutation in GNE. Interestingly, the genetic mutations M712T and V727M are both present in Exon No. 12 of the GNE gene, which encodes the ManNAc kinase in the GNE protein. Being in close association, these two mutations are reported to induce a similar effect in HIBM patients [25,26,27]. In the GSE12648 study, samples were taken from muscle specimens (deltoid, biceps, quadriceps, tibialis). Ages of patients ranged between 20 and 59. An additional 10 matched samples were taken from healthy control individuals (deltoid, biceps, quadriceps, gluteus, paraspinal, and triceps muscles), aged 18 to 74. Using these data, GSEA analysis was done for the identification of altered pathways in patients with high COL6A3 vs. low COL6A3 expression levels. An enrichment score was calculated by starting with the most highly ranked genes (according to the gene-level log2 fold changes values), increasing the score when a gene was in the pathway, and decreasing the score when a gene was not in the pathway. The final data were represented as a bubble plot to represent altered pathways among these two groups.

### 2.13. String Analysis

STRING is a well-recognized database of known and predicted protein–protein interactions. This includes direct (physical) interactions or indirect (functional) associations; they stem from computational prediction and/or knowledge transfer between organisms and/or from the interactions aggregated from other (primary) databases. The string analysis was included to find the interacting partners between key genes identified in this study.

### 2.14. GEO Data Analysis 

The published literature and GEO database were also searched to identify high-throughput-based studies reporting altered genes and pathways in HIBM patients. For this, a recent study by Benyamini H et al. showing the transcriptome of kidneys of sick and healthy GNEM743T/M743T mice were also analyzed to check the status of COL6a3 and coexpressed genes in HIBM-specific diseased models (GEO database GSE141302) [28]. For expression analysis in human muscular disorder–specific tissues, patient data for Tibial muscular dystrophy, Polymyositis, Necrotizing Myopathy, Dermatomyositis, Inclusion-Body Myositis, and Non-specific myositis and respective control sets were analyzed from the GSE128470 and GSE42806 datasets.

### 2.15. Statistical Analysis

All in vitro experiments were performed at least in triplicate. A two-tailed Student’s *t*-test was applied for the calculation of *p*-values using Microsoft Excel. Data with *p*-values < 0.05 were considered significant. Wherever applicable, error bars were used to denote standard deviation.

## 3. Results

### 3.1. Differential Expression of Various Noncoding RNAs Are Found to Be Associated with GNE V727M Mutation

Analysis of the RNA-sequencing profile in GNE V727M mutant overexpressing HEK293 cells as compared to WT overexpressing HEK293 cells showed a total of 301 protein-coding genes to be differentially expressed. Of these, 233 mRNAs were upregulated (fold change >2; *p*-value < 0.05), and 67 mRNAs were significantly down-regulated (fold change <2; *p*-value < 0.05) in GNE V727M mutant overexpressing cells compared to WT overexpressing HEK293 cells (Appendix A). A total of 572 noncoding genes were also found differentially expressed (fold change >2 or <2; *p*-value < 0.05) between GNE V727M mutant HEK293 cells and GNE WT overexpressing cells (Figure 1a). Few of the targets enriched in the RNAseq study were further evaluated by quantitative real-time PCR data, which also showed a consistent trend with the results (Appendix A). Interestingly, apart from coding genes, many noncoding RNAs were also found enriched, which included many lincRNAs, snoRNAs, and miRNAs (Figure 1b). Since the role of ncRNAs in affecting various cellular processes remains largely elusive, the potential role of these ncRNAs was analyzed in HIBM myopathy.

### 3.2. lincRNAs Are the Most Highly Deregulated Group by the GNE V727M Mutation

Since lincRNAs are one of the interesting and emerging long noncoding RNA groups enriched in our study but have been the least explored in HIBM myopathies, this group of noncoding RNAs was selected for further analysis. Our preliminary investigation showed a majority of the identified lincRNAs in our study were located on Chromosome 1 (n − 6), 17 (n − 6), and 7 (n − 5) (Figure 1c). Interestingly, more than 91% of the identified lincRNAs had a down-regulation trend (n43/47) (Figure 1d, Appendix A). 

### 3.3. LincRNA Target Analysis Showed Their Role in the Regulation of Wnt Signaling, TGF β Pathway, and Apoptotic Signaling

To predict the function of differentially expressed lincRNAs, all the 47 lincRNAs identified were selected for downstream target analysis. For this, lincRNAs were subjected to target prediction using rtool (http://rtools.cbrc.jp/cgi-bin/RNARNA/index.pl, accessed on 26 May 2021), using the minimum energy-based ranking method as the target selection criteria (Figure 2a). For each lincRNA, a maximum of the top 400 targets were selected for analysis, which finally led to the identification of a total of 8416 unique targets for 47 lincRNAs. To identify the most important targets, only the targets that were found to be regulated by more than or equal to four of the lincRNAs were shortlisted for downstream studies. This led to the identification of a total of 839 targets (Appendix A). Since transcription factors could specifically regulate lincRNA-associated genes, we first predicted various transcription factors that could be regulating these 839 target genes. Our analysis of potential factors using the Gencodis tool suggested a possible regulation by STAT1, SP1, TP53, and MYC transcription factors (Figure 2b). To identify the key genes with an important role, we next used panther analysis to find key pathways in which the lincRNA target genes were involved. Our study showed targets of the lincRNAs were part of essential processes of WNT signaling, TGF-β, and Apoptotic inducing pathways (Figure 2c,d); some of these pathways are well known to be important in various myopathies and dystrophies [29,30,31,32,33,34,35].

### 3.4. Collagen VI Family Member COL6a3 Is a Key Target Deregulated via GNE V727M Mutation

In order to find the key genes affected by V727M mutation, we first checked for the lincRNA specific targets that were also found differentially expressed in our RNAseq studies. Interestingly, we found 13 such molecules common between 839 lincRNAs targets that were also reported differentially expressed in our RNAseq-based studies (Figure 3a, Appendix A). GO cellular component analysis showed these targets to be involved in extracellular matrix-specific functions (Figure 3b). Further, the Medgen database (https://www.ncbi.nlm.nih.gov/medgen acessed on 4 November 2021) was used to identify critical targets reported to be associated with diseases. Three of the coding genes (RCAP, RP11-83N9.5, and COL6a3) were found to be associated with floating harbor syndrome, retinitis pigmentosa, and myopathy disorder, respectively (Figure 3c). Among these, collagen family proteins were an interesting target, as collagen family members are deregulated in a large group of myopathies [36,37,38,39,40]. A large set of collagen family proteins were found deregulated in our expression data. Similar to our findings, a recent study by Chakrovarty S et al. also analyzed primary myoblast tissue carrying the V727M mutation for expression of altered genes using a high throughput expression profiling approach. Interestingly, they also reported COL6A3 to be a key gene upregulated (more than 2.5 fold) in HIBM patients [41]. Since both the studies confirm COL6A3 levels to be altered in HIBM patients harboring the V727M mutation, it was selected for further analysis, as it may induce essential signaling deregulation in HIBM myopathy.

### 3.5. HIBM Patients with Varying COL6A3 Expression Show Alteration of Myogenic-Associated Pathways

To identify the key genes altered in HIBM patients, the GEO datasets with the diffrential expression profile were searched. Only one effective study in patient samples was identified, including HIBM patients carrying the M712T Persian Jewish founder mutation in GNE and presenting with mild histological changes. This study was selected, as M712T and V727M are present in the same exonic region that encodes the ManNAc kinase in the GNE protein. Further, being in close association, the impact of both these mutations is reported to induce a similar physiological effect in patients [25,26,27]. In our study, these patient samples were divided on the basis of expression of COL6a3, with those above the median as highly expressed and those below the median as lowly expressed. Finally, the GSEA enrichment of pathways in the two groups was analyzed. Interestingly, we found the epithelial-to-mesenchymal transition pathway to provide the most enrichment in the COL6A3 high-expressing group. Additionally, the genes specific to myogenesis, hypoxia, apoptosis, and the apical junction were also significantly enriched in these groups (Figure 4a,b). Thus, the patients with high COL6a3 expression in HIBM myopathy seem to have an enrichment of pathways that affects muscle cell formation, cell development, and cell death. Interestingly epithelial-to-mesenchymal transition was an important pathway in this group, though it has been little explored in HIBM myopathy.

### 3.6. COL6a3 Associated 13 Gene Signature Is Highly Deregulated by GNE V727M Mutation

To further explore the role of the COL6a3 gene in inducing GNE myopathy, we investigated the COL6a3 coexpressing gene sets. For this, COL6a3 coexpressing proteins were identified using the “geneslikeme” tool. To select stringent targets, only top targets were selected for downstream analysis (Appendix A). Interestingly similar to COL6a3 expression, the RNAseq study showed around 20% (12/60) of the coexpressed targets showing a similar overexpressed pattern in GNE-specific V727M mutation cells in comparison to WT form, whereas none of the 60 targets were found to show a down-regulated trend (Figure 5a,b, Appendix A). RNAseq analysis results show COL1a2, DCN, MFAP5, and LUM to be the most overexpressed targets, whereas CCDC80 and CDH11 were among the least altered targets (Figure 5b, Appendix A). Interestingly, similar to COL6A3-associated enriched pathways as per GSEA analysis, most of these genes were also involved in the EMT pathway, and some of these genes involved apoptotic and hypoxia signaling (Appendix A). To validate the importance of these genes, we also investigated reported HIBM-specific NGS data that focused on the effects of HIBM-specific GNE mutation. For this, the transcriptome analysis of kidneys of the sick and healthy GNE M743T/M743T mice reported by Benyamini H et al. (GEO database GSE141302) was investigated, as such mice usually die a few days after birth from severe renal failure. Thus, the status of COL6a3 and the coexpressed genes in HIBM-specific disease models were analyzed in the kidneys of mice (Figure 5c) [28]. Interestingly, our analysis of the data of Benyamini H et al. clearly showed a highly significant upregulated profile of 11 (out of 13 genes of interest) genes in the kidney tissue of sick HIBM mice models vs. healthy cases (as well as in 10/12 sick HIBM mice models vs. wt cases) (Figure 5c).

### 3.7. Multiple Musculoskeletal Disorders also Show a Deregulated Trend for the Identified 13 Gene Signature

In order to determine the importance of selected genes in various muscular disorders, we checked the Gene Expression Omnibus database for submitted data using the “myopathy” term. The data submitted to the GEO database was filtered to find studies that included myopathy, either alone or in combination with other diseases that affect the physiological function of muscles. These results were further shortlisted using filters including “expression profiling by array” “homo sapiens,” and “tissue” as key filters. These were finally checked manually for samples with various muscular disorders. Based on the above criteria, we identified GSE-128470 and GSE-42806 as two important datasets that have microarray expression profiling data available for various muscular disease-specific samples and controls. This also included studies by Greenberg SA et al., who included patients of Necrotizing myopathy as well as Polymyositis, Inclusion-Body Myositis, Dermatomyositis, and Non-specific myositis. Since all data represented a human disease that causes muscle weakness affecting various body parts, we also covered these in our analysis by looking at COL6A3 and 10 associated genes. Interestingly, similar to our findings, these datasets clearly showed a significant upregulation of COL6A3 as well as the majority of coexpressed genes in such muscular diseases (Figure 6). Thus, altered expression of COL6A3 and the coexpressed gene signature identified in our study is not only limited to deregulation in HIBM myopathy but may also seem to play an equally essential role in inducing other muscular disorders. Further, we looked for deregulation of Col6A3 in GNE myopathy in comparison to the different myositis data. The expression of COL6A3 in V727M mutation–harboring GNE patients was higher than 2.5-fold in comparison to all myositis (Polymyositis, Inclusion-Body Myositis, Dermatomyositis, and Non-specific myositis). This suggests that the upregulated expression of COL6A3 is greater in myopathy patients harboring GNE-specific V727 mutation compared to other muscular diseases.

### 3.8. PAX2, STAT2, and LHX4 Seem to Be the Key Players in the Regulation of the 13 Gene Signatures

Further, to predict potential transcription factor binding sites that could be regulating this set of genes, we predicted key potential transcription factors (TFs) that could be binding at the promoter of these genes. For this, we used the CiiiDER tool and predicted TFs binding regions within 3000 bp upstream in COL6a3 and the coexpressed gene sets (Figure 7a). Interestingly, we found PAX2, STAT2, and LHX4 were among some of the key transcription factors whose binding sites were frequently present in most of these genes, thereby suggesting a possible co-regulated expression of these genes that requires further investigation. Additionally, string data analysis also suggested that the majority of these coexpressed targets could be interrelated and thus could be responsible for regulating crucial signaling molecules, leading to HIBM-specific pathological outcomes (Figure 7b). To elucidate their predictive role, GO biological pathways analysis was performed, which also showed the importance of these targets in important processes, including extracellular matrix organization, cell adhesion, and skeletal muscle tissue development (Figure 7c). Thus, we conclude that alteration of lincRNAs via V727M mutation could alter COL6a3-associated genes, thereby inducing the HIBM myopathy-associated phenotype, which may require further exploration.

## 4. Discussion

Mutations in the GNE gene are considered to be one of the most common features reported in all HIBM patients and are also shown correlated with altered signaling in such cases. However, little is known about the association between GNE mutations and the noncoding RNAs. Thus, investigating this relationship may help in further understanding the associated modulations induced by the GNE-specific mutations in the development and progression of this disease. Therefore, in this study, we explored the effect of GNE-specific ethnic founder mutation (V727M). This study’s primary goal included identifying deregulated noncoding RNAs, with special emphasis on lincRNAs and the associated targets. For this, the genome-wide RNA sequencing-based expression profiling was done in GNE V727M mutant overexpressing HEK293 cells as well as the GNE wild-type overexpressing HEK293 cells. The lincRNAs with statistically significant differential expression were identified and examined for their role in the HIBM myopathy. As a result, a total of 43 down-regulated lincRNAs and four upregulated lincRNAs were identified in the GNE V727M mutation when compared to GNE wt-expressing HEk293 cells. The biological role of the lincRNAs is complicated and subject to a variety of complex and diverse functions. Recent studies have also shown that many lincRNAs are involved in regulating gene expression by serving as a guiding, scaffolding, signaling, and decoying molecule.

It was initially suggested that lincRNAs act via their interaction with proteins; however, many recent studies have also shown a link between lincRNA direct interaction with sequence-specific RNA targets [42,43,44]. Zealy et al. (2018) showed that the sequence complementarity of lncRNAs was found to show the quantity of target RNA expression being influenced. Interestingly, a positive correlation between lncRNA and target RNAs was predicted in this study, which was also validated by the group using various wet-lab experiments (including RNA affinity pull-down, microarray, and RNA-sequencing analysis). Similarly, Gong C et al. also experimentally showed lincRNAs directly base-pair with RNAs and subsequently recruit proteins involved in their degradation. Moreover, leads from studies like Carrieri, C. et al. further show that the base pairing of lincRNAs in *trans* is crucial for loading mRNAs on active polyribosomes and enhancing the translation of such sequence-specific mRNA targets. Thus, to access the functions of the identified lincRNAs, the prediction and analysis of the potential target mRNAs regulated by the lincRNAs was selected as a reliable method. Next, the identified targets were used to identify and categorize key transcription factors that could interact with lincRNAs and may be involved in regulation. Analysis of these targets showed STAT1, SP1, and MYC to be some of these genes’ key transcription factor regulators. STAT1 is an important transcription factor that has been reported to regulate the expression of important genes controlling cell growth, differentiation, apoptosis, and immune functions in various diseases [45]. Phosphorylated forms of STAT1 homo- and heterodimer forms are known to activate gene expression. Sp1 is another important transcription factor that is reported to regulate genes involved in essential cellular functions, including proliferation, differentiation, DNA damage response, apoptosis, senescence, and angiogenesis [46,47,48,49,50,51,52,53]. c-Myc, on the other hand, is also a critical transcription factor that behaves as a global regulator of transcription. Many genes for cell cycle regulation, metabolism, ribosome biogenesis, protein synthesis, and mitochondrial function are over-represented in the c-Myc target gene network [54]. Thus, our analysis suggests that altered expression of targets reportedly associated with these transcription factors could also be playing an essential role in myopathies. Further, to investigate the integrated role of these lincRNA targets, panther pathway analysis was performed for elucidating their potential functions. Our study revealed WNT, TGF-β, and apoptotic signaling pathway specific genes could be enriched due to altered lincRNA expression. Wnt pathways have been shown to play an important role in the formation of muscle fibers during prenatal and postnatal myogenesis, and this has been correlated with various skeletal muscle diseases [55]. Furthermore, some of the Wnt signaling genes, such as Phosphoinositide-phospholipase C β1 (PLCβ1), are reported to play a crucial role in initiating the genetic program responsible for muscle differentiation and have been correlated with myotonic dystrophies [56]. Further, TGF-β pathways have been reported to alter in various myopathies. Recently, it has been shown that inhibiting TGF-β can reduce fibrosis and improve regeneration in several genetic forms of myopathy, including dystrophin-negative Duchenne muscular dystrophy and Marfan syndrome [57,58,59]. These results suggest that GNE V727M induced deregulation of lincRNAs may cause myopathic effects by modulating WNT signaling, TGF-β, and apoptotic signaling pathways.

To further identify the key genes affected by lincRNA deregulation, we correlated the above targets with differentially expressed genes identified in our study. Interestingly, COL6a3 was an essential target that has been reported to be altered in multiple myopathies [60,61,62,63,64]. Mutations in COL6A3 cause a spectrum of muscle diseases, from Bethlem myopathy at the mild end to the severe Ullrich congenital muscular dystrophy [65]. COL6a3 deficiency seems to alter extracellular matrix structure and biomechanical properties and could lead to increased apoptosis and oxidative stress, decreased autophagy, and impaired muscle regeneration [65]. Since COL63 and other collagen family members have been the key candidates in various myopathies, we suggest they could play an important role in inducing HIBM. For this, we checked the status of genes coexpressed with COL6a3 and their expression in our study. Interestingly, a large group of COL6a3 coexpressing genes was not only found altered in our analysis but, similar to COL6a3, all of these showed up-regulatedexpression.

To analyze if the trend observed for COL6a3 and coexpressed genes was due to GNE-specific mutations, we analyzed another high throughput NGS-based study reported by the Benyamini et al. This group checked knock-in mice carrying GneM743T/M743T, one of the most frequent mutations in GNE myopathy patients. Further, to study the expression status of our genes in HIBM myopathy, we specifically analyzed the kidney tissue-specific transcriptome of sick and healthy mice, as the majority of the mutated mouse usually die a few days after birth from severe renal failure. Interestingly, analysis of the data of Benyamini H et al. clearly showed a highly significant upregulated profile of 11 genes in the kidney tissue of sick HIBM mice models vs. healthy cases. This suggests that various HIBM myopathy-specific GNE mutations could induce downstream pathogenic effects by deregulating the COL6a3 and coexpressed gene signature in HIBM myopathy.

To further validate if the observed trend for COL6A3-associated 13 gene signature played any role in other musculoskeletal disorders, we checked the expression of these gene sets in patient expression profile data using the GEO database. Interestingly, analysis of available patient datasets for more than six muscular disorders from GEO strengthened our trend of deregulated COL6A3 and coexpressed gene axis in many other myopathies and dystrophies. Among these, COL6A3, CDH11, COL1A2, IGFBP4, and THSB2 seem to be the most commonly deregulated genes in terms of expression in these disorders. Since such muscular disorders are majorly associated with dysfunction of muscle fibers, we expect deregulation of COL6A3 and associated gene sets may be essential in affecting muscular functioning and cell death. Further studies are required in order to investigate the individual and combinatorial effect of the deregulation of these sets of genes in muscular disorders.

Since these 13 gene sets are mostly associated with a similar upregulated trend in diseases, we hypothesized their induced coregulation in muscular disorders. Interestingly, we found these genes could have a common regulation by various transcription factors (TFs), such as like STAT2, PAX2, and LHX4. Moreover, all of these targets were over-expressed; thus activation of these was possibly due to myopathy-specific mutations leading to the disease phenotype, which requires further investigation. Further, each of these TFs is also correlated with other muscular diseases. DCN is one such coexpressed gene that plays an essential role in collagen fibril assembly and has also been found to be associated with multiple cell surface receptors. It is also reported to associate with the glycosylation process and has been associated with corneal dystrophies [66]. Similarly, Lumican (LUM) is another coexpressed gene that may regulate collagen fibril organization extracellular matrix structural constitution, whereas THBS2 is another important gene that encodes a homotrimeric glycoprotein that mediates cell-to-cell and cell-to-matrix interactions [67]. Additionally, GO biological pathway analysis of the COL6A3 and coexpressed genes show these to be involved in extracellular matrix organization, cell adhesion, and skeletal muscle development. Thus, our analysis strongly suggests altered signaling via COL6a3-associated signaling, which requires further analysis.

Overall, the present study revealed that GNE V727 mutation significantly altered lincRNAs. This also indicates that altered lincRNAs may mediate the interaction between molecules, further implicating the GNE mutation role in the regulation of key pathways, thereby leading to myopathy. We conclude that HIBM mutation-induced lincRNAs could play a key role in the induction of HIBM myopathy by regulating the expression of COL6a3-associated genes (Figure 7d). It is noteworthy to further explore the key mechanism and functional role of these lincRNAs modulated by the GNE V727M mutation. Moreover, further experimental support may be needed to narrow down to some key lincRNAs that could be affecting the signaling pathways in myopathies. Further research aimed to address the functional significance of other GNE mutations in altering lincRNAs and associated signaling pathways would be helpful to elucidate the common mechanism responsible for muscle degeneration. Additionally, analysis of various technologies investigating RNA–protein interaction could be beneficial to explore the molecular role of GNEV727M mutation in altering lincRNAs.

Thus, our study is an approach to reveal an altered profile due to GNE-specific V727m mutation using in vitro models. Although the metabolic importance of GNE is well reported, the role of associated key mutations is unknown. Thus, this study aims to provide novel insights into the role of lincRNAs affected due to V727M mutation in the GNE gene. Interestingly, a highly correlated expression pattern of COL6a3 and the co-associated gene signature was observed, which was also found to be differentially expressed in musculoskeletal disorder-specific profiles. These findings could play an important role in understanding the molecular effects induced by GNE mutations. Moreover, further studies will be helpful to explore the role as well as to define the functions of the COL6a3-associated gene signature to understand the molecular mechanism leading to myopathies. Further, targeting such pathways could potentially be an effective method for treating HIMB and other myopathies.

## 5. Conclusions

The present study describes the GNE V727M–specific associated coexpressed gene network in HIBM myopathy. Our results suggest that a 13 gene signature (COL6A3, COL1A2, MFAP5, DCN, PDGFRB, LUM, IGFBP4, EMILIN1, CCDC80, FAP, CDH11, THBS2, and ELN) is altered in in vitro and in vivo HIBM models as well as in many other muscular disorders. Thus, they seem to play an important role in the deregulation of many important cellular pathways relevant to muscle development and functions. Overall this novel COL6A3-specific 13 gene signature altered in HIBM and multiple myopathies is not only functionally important but equally holds essential diagnostic as well as therapeutic importance that needs further investigation.

## Figures and Tables

**Figure 1 genes-14-00567-f001:**
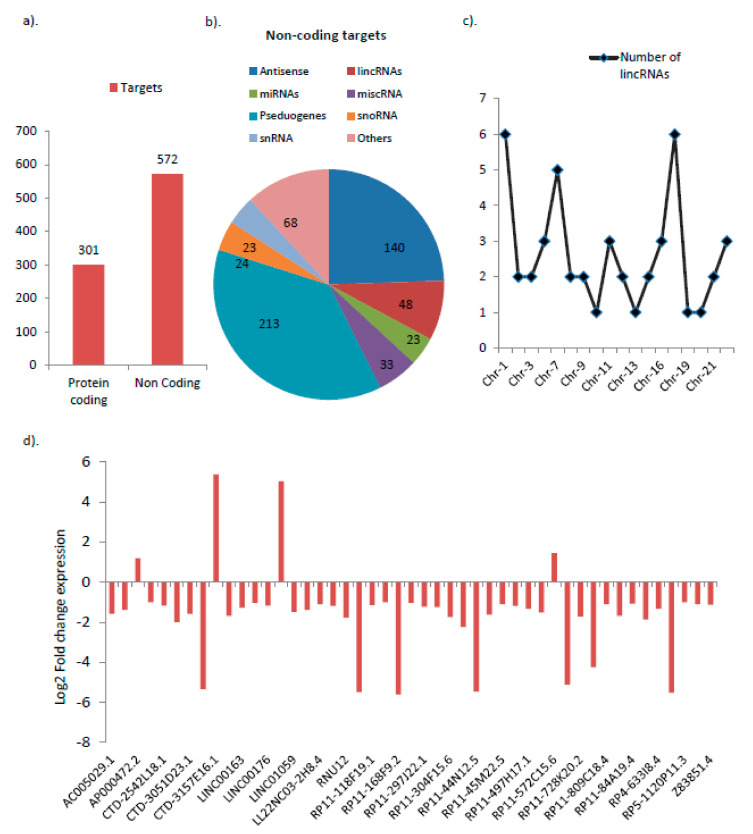
RNA sequencing results in GNE mutant cells compared wildtype overexpressing cells: (**a**) Coding and non-coding targets identified. (**b**) List of key non-coding targets identified (**c**) Chromosome specific distribution of lincRNAs. (**d**) Log2 fold change expression values of lincRNAs as per RNAseq data.

**Figure 2 genes-14-00567-f002:**
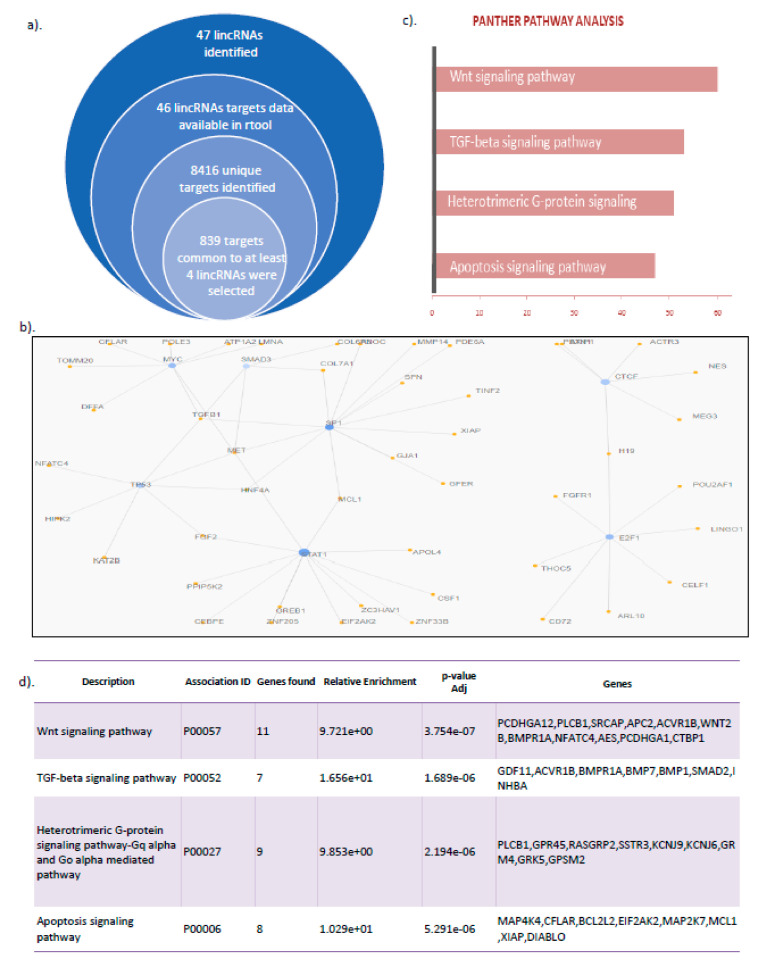
LincRNAs target analysis approach: (**a**) Selection criteria followed. **(b**) Identification of potential transcription factor regulators of selected genes. (**c**) Top pathways affected (**d**) Gene-specific to each pathway.

**Figure 3 genes-14-00567-f003:**
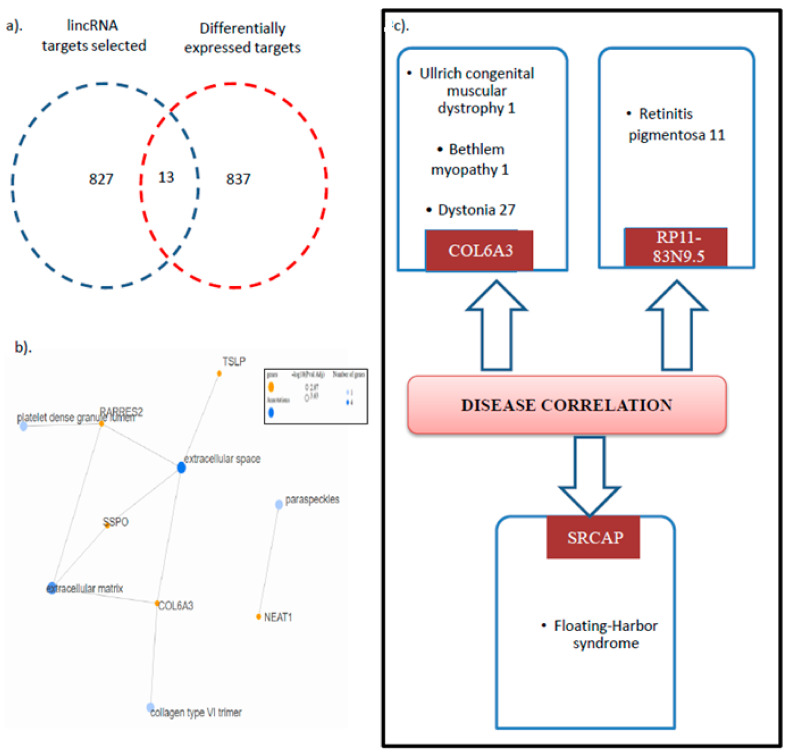
Identification of key targets: (**a**) Venn diagram to show common lincRNAs targets along with significant differential expression; (**b**) GO cellular compartment analysis of identified targets; (**c**) Medgen database supported disease association of crucial targets.

**Figure 4 genes-14-00567-f004:**
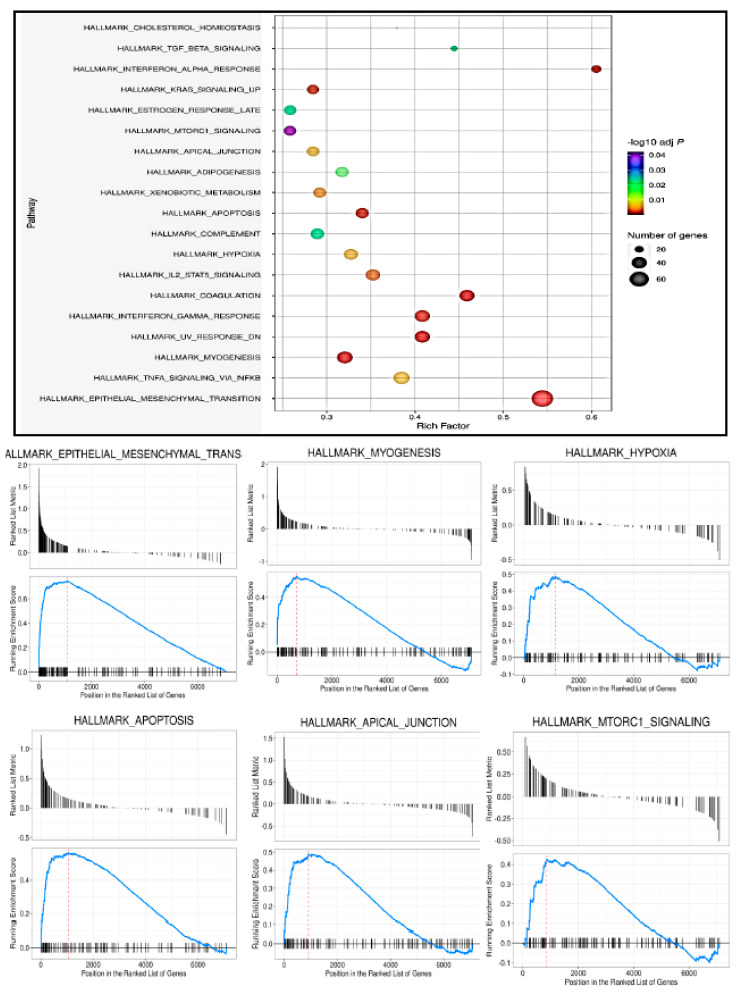
GSEA analysis of Pathways deregulated in HIBM patients with differential COL6A3 levels. Bubble plot to show significant pathways in patient with differential COL6A3 levels. The red dashed line indicates the enrichment score, which is the maximum deviation from zero. Key pathways found enriched in analysis.

**Figure 5 genes-14-00567-f005:**
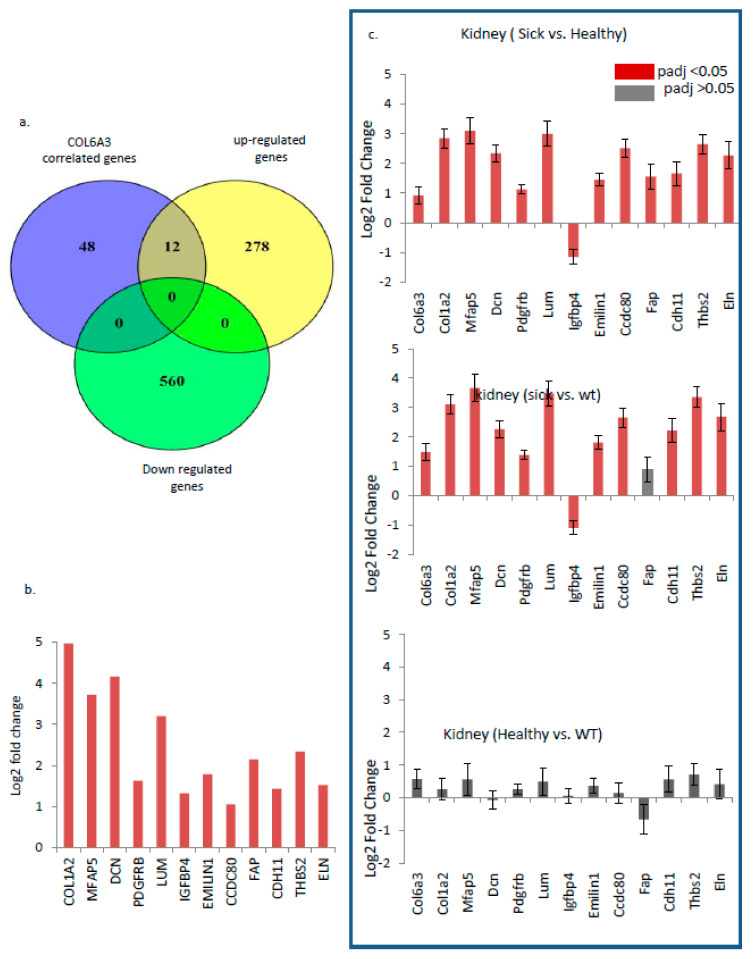
Analysis of COL6a3 correlated gene using various NGS data: (**a**) Venn diagram to show upregulation and downregulation status of COL6a3 co-expressed genes. (**b**) Log2 fold change expression of COL6a3 co-expressed genes (in RNAseq data). (**c**) Status of our COL6a3 and co-expressed 13 genes in HIBM specific diseased models in kidney tissue (GEO database GSE141302) in (i) Sick HIBM mice models vs. healthy mice (ii) Sick HIBM mice models vs wt mice, (iii) Healthy mice vs WT mice.

**Figure 6 genes-14-00567-f006:**
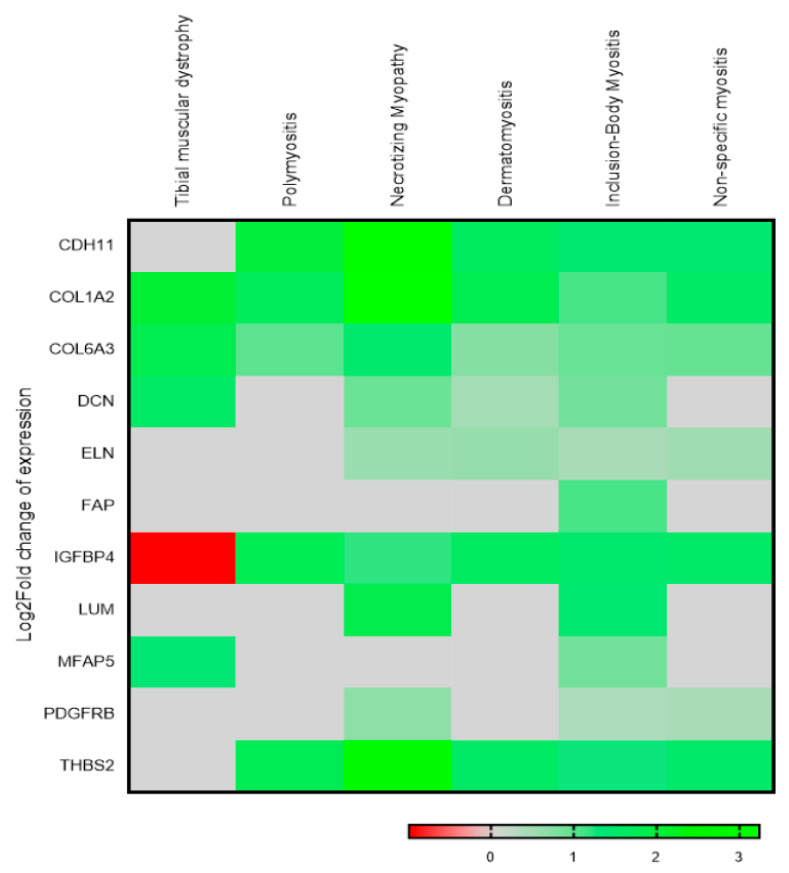
Heat map to show altered expression of COL6A3 and identified 10 co-expressed genes in various muscular disorders.

**Figure 7 genes-14-00567-f007:**
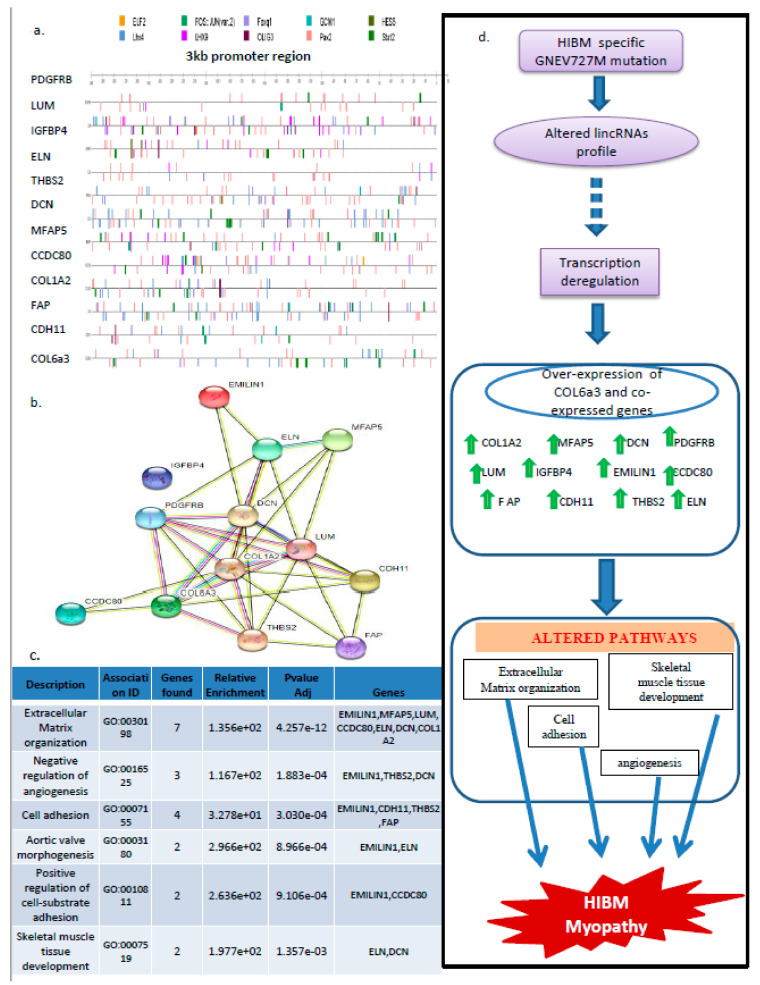
Functional role of COL6a3 correlated gens in HIBM (**a**) Key potential transcription factor binding sites within 3 kbp upstream sequences of COL6a3 and co-expressed genes, as identified by CiiiDER tool (**b**) String analysis based interaction map of COL6a3 and co-expressed targets. (**c**) GO Biological pathways analysis of COL6a3 coexpressed genes. (**d**) Cartoon representation summarizing highlights of the altered GNE V727M/ lincRNAs/ COL6a3 co-expressed genes axis identified in this study.

## Data Availability

The data supporting this study’s findings are available from the corresponding author upon reasonable request.

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
