# Peer review of "Genetic Analysis of HIBM Myopathy-Specific GNE V727M Hotspot Mutation Identifies a Novel COL6A3 Allied Gene Signature That Is Also Deregulated in Multiple Neuromuscular Diseases and Myopathies"

_genes, 2023, doi:10.3390/genes14030567_

Round 1
Reviewer 1 Report
Attri et al. found a deregulation of different lincRNAs, among which Col6A3 in an heterologous system (HEK293 cells) carrying the V727M sequence variant in the GNE gene. They confirm the relevance of this finding by using GEO expression databases in primary myoblasts isolated from patients affected by hereditary inclusion body myopathy, and by analyzing the transcriptome of GneM743T/M743T mutated mice.
The work needs major revisions. The deregulation of Col6A3 needs to be confirmed in primary myoblasts or muscle cell lines carrying the V727M GNE mutation. The description of the analysis of the GEO expression database from patients is not clear to the reader. How many patients were analyzed? The all had the Mutation M712T? The reader has no information of the clinical impact of this mutation in respect to the V727M.
In the analysis of other muscle diseases, all the data come from myositis. The only hereditary myopathy is tibial muscular dystrophy. It is interesting that this molecule is deregulated in myositis. However, the Authors wrote that they searched using the term "myopathy". Therefore the process of analysis is not clear. What did they found in other muscular dystrophies? How high was the deregulation of Col6A3 in GNE Myopathy in comparison to the myositis. The quantification of this difference is not clear.
Reviewer 2 Report
The article describes that genetic analysis of HIBM myopathy specific GNE V727M 2 hotspot mutation identifies a novel COL6A3 allied gene signature that is also deregulated in multiple neuromuscular diseases and myopathies.
The work is an important contribution to the field.
Round 2
Reviewer 1 Report
The Authors ahve answered my questions.